# Insight into the Organization of the B10v3 Cucumber Genome by Integration of Biological and Bioinformatic Data

**DOI:** 10.3390/ijms24044011

**Published:** 2023-02-16

**Authors:** Szymon Turek, Wojciech Pląder, Yoshikazu Hoshi, Agnieszka Skarzyńska, Magdalena Pawełkowicz

**Affiliations:** 1Department of Plant Genetics, Breeding and Biotechnology, Institute of Biology, Warsaw University of Life Sciences, 02-776 Warsaw, Poland; 2Laboratory of Plant Environment Science, Department of Agriculture, School of Agriculture, Tokai University, 9-1-1 Toroku, Higashi-ku, Kumamoto 862-8652, Japan

**Keywords:** cucumber genome, comparative genomics, chromosome rearrangements, repetitive elements

## Abstract

The availability of a well-organized and annotated reference genome is essential for genome research and the analysis of re-sequencing approaches. The B10v3 cucumber (*Cucumis sativus* L.) reference genome has been sequenced and assembled into 8035 contigs, a small fraction of which have been mapped to individual chromosomes. Currently, bioinformatics methods based on comparative homology have made it possible to re-order the sequenced contigs by mapping them to the reference genomes. The B10v3 genome (North-European, Borszczagowski line) was rearranged against the genomes of cucumber 9930 (‘Chinese Long’ line) and Gy14 (North American line). Furthermore, a better insight into the organization of the B10v3 genome was obtained by integrating the data available in the literature on the assignment of contigs to chromosomes in the B10v3 genome with the results of the bioinformatic analysis. The combination of information on the markers used in the assembly of the B10v3 genome and the results of FISH and DArT-seq experiments confirmed the reliability of the in silico assignment. Approximately 98% of the protein-coding genes within the chromosomes were assigned and a significant proportion of the repetitive fragments in the sequenced B10v3 genome were identified using the RagTag programme. In addition, BLAST analyses provided comparative information between the B10v3 genome and the 9930 and Gy14 data sets. This revealed both similarities and differences in the functional proteins found between the coding sequences region in the genomes. This study contributes to better knowledge and understanding of cucumber genome line B10v3.

## 1. Introduction

The cognitive process of genomes is being refined constantly, and newer versions of sequenced genomes are being produced based on progressively more advanced sequencing techniques. In the *Cucurbitaceae* family, many genomes have already been obtained, belonging to the tribes *Benincaseae*, *Cucurbiteae*, *Sicyoeae*, *Momordiceae*, and *Siraitieae*. These genomes vary in size from 204.8 to 919.76 Mb and in the genomic draft arrangements in chromosomal structures, while they share common features encoded by genes [1]. Having a genome of adequate assembly quality allows for resequencing related analyses and provides a suitable reference point for comparative analyses. For the reference genome of cucumber *Cucumis sativus*, several genomes have been described previously.

Among the most well-known and described reference lines of *Cucumis sativus* are 9930 [2], a Chinese line, and Gy14 [3], a North American line. The sequencing of the European version of the *Cucumis sativus* genome was described in a publication about assembly of B10v3 genome [4], where researchers managed to obtain a complex genome of size 342,288,160 bp and 8035 contigs. Although the B10v3 genome is the longest cucumber genome, it has a limited number of contigs assigned to chromosomes.

Recently, the graph-based pan-genome for *Cucumis sativus* was constructed from 12 cucumber line sequences [5], where further studies revealed structural, functional, and sequence variations related to agronomic traits and domestication. The resulting pan-genome combines information on cucumber genomes assembled only at the chromosome level, indicating the need for a well-described reference both at the structural level of identified genes and at a larger structural level such as chromosomes.

For the B10v3 genome, part of the contigs were assigned to the corresponding chromosomes as a result of experimental studies. A genome-wide study of B10v3 [4] described the use of markers to target contigs to individual chromosomes. FISH-BAC analysis using clones from a cucumber BAC library [6] was also conducted, whereby chromosome-specific BACs were identified but not aligned to B10v3 version. Moreover, the DArT-seq analysis [7] has been completed for the B10 genome and was used for the assignment of contigs to chromosomes in the B10v2 version [8].

Today, advances in the development of bioinformatics programs make possible the alignment and arrangement of contigs after sequencing against selected reference genomes. Over the past few years, programs for this purpose have made significant advances in both the accuracy of the alignment and the time required, enabling simple organization of contigs in genomes. These programs also cope well with plant genomes, which have many repetitive elements and mobile fragments. Thus, it became possible to use bioinformatics tools to organize the contigs in the B10v3 genome and obtain in silico assignment of contigs to chromosomes. The use of such software offers the possibility of creating appropriate genomic assemblies to enable subsequent comparative analyses, and also the construction of pangenomes, as described in a publication on the creation of the *Aspergillus fumigatus* pangenome [9].

In addition, more and more information on sequenced cucumber genomes is available in reference databases. This creates opportunities to compare genomes and detect similarities and differences between them. It is possible to integrate bioinformatics data with the results of laboratory experiments, providing confirmation of the information obtained in silico and vice versa.

In the present work, the objectives of the analyses were (1) to compare the cucumber genomes (B10v3, 9930, Gy14) and arrange B10v3 contigs into chromosome structure using bioinformatics tools based on comparative homology, (2) to compare the accuracy of in silico tools in plant genome analysis, (3) identification of similarities and differences at the level of encoded protein sequences between the B10v3, 9930 and Gy14 genomes, and (4) data integration from genetic, molecular, and cytogenetic analyses reported in the literature with bioinformatic results of contig ordination.

Here, we present results for individual contigs of the B10v3 genome sequenced and assigned to chromosomes using in silico tools. We also preface the functional annotation results by presenting a comparison of the proteins encoded by the identified structures on the B10v3, Gy14, and 9930 genomes, in which high similarity of the encoded amino acid sequences were found. In addition, the integration of previously obtained data (FISH, DArT-seq and other marker sets) confirmed the reliability of the results obtained in the bioinformatics analysis, ordering the information on the distribution of contigs in the B10v3 genome. The use of computer-based techniques and laboratory experiments is in the mainstream of recent concepts on genome assembly [10], providing a deeper understanding of the studied genomes. Using the chosen software and a set of results from laboratory experiments, it was possible to organize the already complex B10v3 cucumber genome to a level that had never been achieved before.

Finally, in this work, we present a new pipeline for comparative genomic analyses integrating high-throughput sequence data with biological data, resulting in the consolidation and structuring of knowledge on genomic draft structure (Figure 1).

The results presented in this work are of great importance for biologists and bioinformaticians who have to deal with a variety of data resources in cucumber, but also in other plants. The relevance of the work lies in the presentation of an appropriate approach for the analysis of experimental data and different genomic data, showing the pros and cons of currently-used algorithms for data integration. It also unifies the used descriptions of the cucumber chromosome structure and explains the discrepancies between the different versions of the cucumber genome. Based on our approach, we identify trends that need to be addressed for efficient comparative analysis of high-throughput data, taking into account the obstacles we encountered when using current genome comparison tools. The future will bring ever increasing resources of information. It is important that these data can be correlated and used as efficiently as possible.

## 2. Results and Discussion

### 2.1. Comparison of Genomes Structure

Individual metrics for the B10v3, Gy14, and 9930 genomes were generated using the sequence-stats program and are presented in Table 1. The selected reference genomes differ in their degree of assembly, with the B10v3 genome consisting of the largest number of contigs—8035. The 9930 genome constitutes 7 chromosomes and 78 additional scaffolds while the Gy14 genome consists of 7 chromosomes and an additional chromosome 0. Comparing the lengths of the individual genomes, the largest is the B10v3 genome, with a length of 342,288,160 bp, followed by the Gy14 genome with 258,626,374 bp, and the 9930 genome with 226,211,662 bp. The percentages of individual bases (A, T, C, and G) in the sequences remain similar, while the difference is the N50 value, which is defined as the length of the shortest contig for which longer and equal length contigs [11] cover at least 50% of the assembly.

### 2.2. Comparison of Sequences on Multiple Sequence Alignment Plot

Analyzing the data statistics (Table 1) for each genome, the differences between the size of the genomes and the length of the contigs can be seen. The estimated size of the cucumber genome according to the literature data on flow cytometry is 367 Mb [12]. The genome that comes closest to this value is the B10v3 genome, while the actual number of genes described on the respective B10v3, Gy14, and 9930 genomes is at a similar level.

The multiple sequence alignment (MSA) presented on Figure 2 illustrates differences and similarities of data from CuGenDBv2 database for the B10v3, 9930, and Gy14 genomes. Vertical red lines indicate chromosome boundaries in the genomes. The individually colored fragments correspond to the assigned fragments that are common to two other genomes, while the dark grey fragments indicate the absence of such assignment, meaning that these fragments are specific to a single genome. All sequences that have not been assigned to individual chromosomes in the 9930 and Gy14 genomes can be seen in the graph as additional scaffolds (at the end of the graph for 9930) or chromosome 0 (beginning of the Gy14 graph). For the B10v3 genome, it can be observed that some of the contigs, especially the largest ones, were appropriately color-assigned to fragments in the other genomes, indicating the similarity of these sequences and their presumed assignment in subsequent genomes. Fragments that were not found in any of the other genomes, but B10v3 genome, make up the largest portion among the genomes compared, indicating unique sequenced fragments present only in the genome B10v3.

### 2.3. Homology Based Tools Comparison

Using programs that rely on comparative homology, it was possible to obtain in silico contig arrangement by comparing sequences to the 9930 and Gy14 genomes. The programs used—RagTag, Mauve Contig Mover, and Multi-CSAR—gave different results, varying in the degree of contig assignment to continuous chromosomes. The program that gave the best results was RagTag, which was the only tool that assigned contigs to seven chromosomes. The alignment results of the other programs did not show contig alignment at this level. The Multi-CSAR program aligned contigs to the level of larger scaffolds, while the result of Contig Mover program was alignment with missing fragments. Each program produced FASTA files containing sequences after contig positioning, as shown in the MSA plot generated in Mauve program and shown on Figure 3. Fragments identified as present in both genomes are color-coded, while unassigned fragments are in the grey area. The presented visualizations of the RagTag program’s operation show the alignment of most of the sequences to a length exceeding 200,000,000 bp. The remaining sequences were found in contigs that have not been assigned to any fragment in the 9930 or Gy14 reference genomes.

### 2.4. RagTag Tool Results

The RagTag program possess a merge function that allows us to perform contig arrangement based on several reference genomes simultaneously. Attempting to use this functionality in this case resulted in a lack of contig alignment at the chromosome level, making it impossible to determine the chromosome affiliation of contigs. This allowed us to assume that the three reference genomes are too distinct from each other to perform this type of alignment using three genomes at once. For this reason, separate contig alignment results from the RagTag software were included in the final results: 9930 vs. B10v3 and Gy14 vs. B10v3 (Figure 4 and Figure 5).

Comparing genome fragments arranged into chromosomes with fragments not arranged into chromosomes in the RagTag program, the proportion of repetitive sequences in these fragments was calculated based on the results of the RepeatMasker software [13], available in the B10v3 genome assembly results database [4]. Repetitive sequences in the fragment of ordered contigs into chromosomes accounted for 31% for B10v3 arrangement against the 9930 genome and 34% for arrangement against the Gy14 genome, while in the fragment of unassigned contigs, repetitive fragments accounted for 95% for both arrangements. In addition, the fragment of contigs assigned to chromosomes contained 20,485 genes for the resultant arrangement relative to 9930 genome and 20,718 genes for the resultant arrangement relative to Gy14 genome. The fragment containing the unassigned contigs included 1229 genes for the arrangement relative to the 9930 genome and 996 genes for the arrangement relative to Gy14 genome. Moreover, among the genes that were in the unassigned contig area, 74% of them encoded lncRNAs for both arrangements, whereas for the fragment of ordered contigs, only 22% for the arrangement relative to 9930 and 24% relative to Gy14 were genes encoding lncRNAs. The results obtained with the RagTag program show differences derived from the selected reference genome; however, the total number of repetitive fragments and assigned genes in the fragments after ordering remains similar for arrangements against the 9930 and the Gy14 genome. In both cases, a significant proportion of protein-coding genes were successfully mapped to chromosomes, and the remaining contigs not assigned to chromosomes contained only 317 protein-coding genes for arrangement relative to the 9930 genome and 256 protein-coding genes for arrangement relative to the Gy14 genome, representing 1.9% and 1.5% of all protein-coding genes, respectively. These results show that more than 98% of all protein-coding genes have been assigned to chromosomes by the RagTag software, and the resulting differences depend only on the references used.

As the final result of the RagTag program, a FASTA file with the reordered contigs was obtained, as well as a file in AGP format that describes the arrangement of contigs in the obtained FASTA file. Based on these results, a final table with the arrangement of each contig depending on the selected reference (9930 or Gy14) was prepared and included in the final summary in HTML format that can be searched, filtered, and exported to Excel and CSV (Appendix A).

### 2.5. BLAST Search Results

In order to find corresponding proteins between B10v3 versus 9930 and Gy14 data sets, 16,099 protein sequences encoded by genes from the B10v3 genome were used, and 16,092 matches were found in the 9930 and 16,086 matches in the Gy14 proteins, respectively (Appendix A).

Using the annotation available for the encoded genes in the CuGenDBv2 database, it was possible to assign chromosomes positions functions and ontologies to the searched proteins in BLASTP results. Assigning the searched proteins to the corresponding annotations from the other genomes allowed the probable allocation of contigs on chromosomes in the B10v3 genome.

A subsequent search for 16,099 B10v3 proteins in the local NR database restricted to the taxon *Cucurbitaceae* yielded 15,923 protein sequences (Appendix A). The following Table 2 shows the number of protein sequences hit within each sequence identity range. At 100% similarity, 9519 proteins were found in the 9930 database, 7422 proteins in the Gy14 base, and 7311 proteins in the 3650 taxon base. In the 95–100% identity range, 5008, 4635, and 6002 proteins were found, respectively. There were significantly fewer proteins in the region with identities below 95% in the three comparisons under analysis (Table 2).

The above comparisons yielded a large number of proteins with a high degree of identity, (in the 90–100% range), the majority of identified proteins being 94% for the 9930-base search, 83% for the Gy14 base search, and 93% for the 3650 taxon restricted base search. Searching for common proteins from the B10v3 coding sequences that were found with 100% identity in the 9930 and Gy14 data reveals 5661 common proteins, while extending the identity range above 90% reveals 13,078 proteins matched in both the 9930 and Gy14 protein coding sequences. For the protein search results in the 9930 line, where identity was 100%, and for Gy14 line, where it was below 90%, there were 1179 such proteins. Only 154 proteins were found in the reverse combination, where the identity was 100% for Gy14 and below 90% for 9930. This result indicates a much higher similarity of the searched proteins in the 9930 line than in Gy14 line compared to B10v3. The proteins found by BLASTP were generally mapped with high degree of identity to the proteins of the 9930 and Gy14 data sets. In addition, the comparison of the annotated functions of the proteins found between the lines also showed high similarity, especially for the sequences found with high identity.

The final summary report (Appendix A) includes the results of BLASTP searches in the 9930 and Gy14 genomes obtained with 100% identity. The proportion of proteins with less than 80% identity represented 4.2%, 11%, and 0.03%, in the 9930, Gy14, and taxon 3650, respectively. For proteins (with identities less than 80%) in the 9930 (671) and Gy14 (1788), the 99 protein sequences with identities greater than 90% were found in the NR database search restricted to the *Cucurbitaceae* taxon 3650. Most of them were cucumber proteins.

BLAST analysis was performed on all NR databases available on the NCBI server without taxon-specific restriction for the 192 proteins that had less than 80% identity in the BLAST result for taxon 3650 (Appendix A). Of the 189 sequences found, 12 proteins had more than 90% identity, and all were related to *Cucumis sativus*.

The above results suggests that additional searches of the NR database reveal sequences in the B10v3 genome that are available for different versions of the cucumber genomes not described in the 9930 and Gy14 databases, generated from the FASTA files from CuGenDBv2 database. The results also indicate that most of the matched proteins, even with low identity, are from the *Cucurbitaceae* family, confirming the correctness of the created annotation of B10v3 genome. In addition, for three proteins, no search results were obtained in the whole NR database. This suggests that despite the vast majority of coding fragment similarity between the cucumber genomes, there are still unique coding fragments in the B10v3 cucumber genome.

### 2.6. Integration of DArTseq Experiment Results

In order to assign DArT-seq data with the genetic map [7] and prior B10v3 markers [4], the BLASTN algorithm was applied, and the above characteristics were assigned against the B10v3 reference genome. The results indicated matches of loci to the genome with at least 90% identity (Appendix A).

In addition, a list of contig assignments to chromosomes in the B10v3 genome was obtained by combining the information on the assignment of loci to chromosomes provided in the publication [7] supplement.

### 2.7. Integration of FISH Experiment Results

Intensive karyomorphological studies of cucumber have been carried out since the late 1960’s [14,15,16,17]. However, due to the small size, poor staining, and similar size of the chromosomes, the quantity of data was very limited until the second half of the 1990s. Our team proposed numbering of cucumber chromosomes based on CMA and DAPI fluorescence staining [18]. Many works (for example Hoshi et al. [19]; Tagashira et al. [20]) used the proposed numbering of cucumber chromosomes. The appearance of subsequent papers describing the localization of tandem repeat sequences (45S rDNA, Types III and IV) for the lines 9930 and Gy14 [3,21,22,23] generated difficulties when comparing the results of different team’s works. They appeared as a consequences of different chromosome numbering. To this end, we performed comparative analyses of the results of the above-presented assemblies and aligned our results with those described for genotypes 9930 and Gy14 (Figure 6).

The Chr1, Ch2, and Chr5 of B10 line numbering (BChr) corresponded to Chr2, Chr4, and Chr1 of Chinese cucumber numbering (CChr), thus did BChr3 and BChr7 to CChr3 and CChr7, respectively. According to the previous reports, B10 has five 45S rDNA locations which contain three major (BChr1, BChr2, and BChr5) and two minor (BChr3 and BChr7) signals. The 45S rDNA of Chinese cucumber had three major (CChr1, CChr2, and CChr4) and two minor (CChr3 and CChr7) signals [21].

A BLASTN results for STCs compared to B10v3 genomic sequence determined the assignment of individual identifiers to specific contigs. The results of the BAC analysis, transcribed to the B10v3 version of the genome, show the assignment of STCs to chromosomes which confirms the correctness of our comparisons. The comparison of these results with the results of the RagTag program coincides with the results of the BAC analysis. The following Table 3 presents the results of the search with the BLASTN and the assignment with the results of the BAC analysis.

The integrated results for bioinformatics and biological data (Table 4) show the assignment of contigs to chromosomes according to the chosen analysis. The table combines the results of contig rearrangement with the RagTag program against the 9930 and Gy14 genomes, the results of the BLASTP search for the 9930 and Gy14 proteins, and the contig assignment for B10v3 markers, DArT-seq, and FISH analysis. This data compilation allows for the easy search for information for selected contigs.

The small inconsistency in regard to FISH results of ctg 2607 may be related to misinterpretation of chromosome identification after hybridization. It can be related to the above-described factors, namely, small chromosome size and their similarity. The rest of our FISH results perfectly matched the chromosomes in 9930 and Gy14 lines. Additional in silico analyses of selected sequences specific for 18S rDNA (NC_026660.2; NC_026658.2; NC_026656.2) identified contig 6107 and 5660 from Gy14 genome and placed them to chromosome 1 and 4. This is in agreement with previously reported results for this gene. One might ask, what about the other three chromosomes for which rDNA probes have been assigned? Taking into account the incomplete assembly of cucumber genomes resulting from the large number of repeated sequences occurring in them, which also include rDNA sequences, it can be assumed that the lack of detection on the remaining chromosomes is the result of their lack of assignment to already defined structures.

Chromosome polymorphism is a well-described phenomenon. It is presented not only among species but also between cultivars [3,19,24]. Here, we compared cucumber breeding lines: B10, 859, and *C. hardwickii* (CHS55) (Figure 7).

The comparison of chromosomes between B10 and 859 showed chromosomal polymorphism on chromosomes 1, 6, and 7. In chromosome 1, the 45S rDNA was located at the long arm of the pericentromeric region of B10, while, for 859, on the pericentromeric region of the short arm. The terminal region of the short arm of B10 chromosome 6 did not possess any heterochromatic bands and Type IV signal. In the case of 859, the terminal region of the long arm neither possessed a heterochromatic band nor a Type IV signal. The most remarkable differences are specific for chromosome 7; the 45S rDNA signals are located at the pericentromeric region in the long arm of B10 and, to the contrary, on the pericentromeric region of the short arm in the 859 line. Moreover, chromosome 7 had the Type IV signals at the pericentromeric and the terminal regions of the short and long arms of the 859 line, while for B10, the signal is located only at the terminal regions of the long arm.

The CHS55 had characteristic banding after DAPI and CMA staining, which makes chromosome identification possible. However, the tandem repeat signal localization showed a large difference between wild species and cultivars. The major 45S rDNA was detected in chromosome 4, and minor signals were detected in chromosomes 2, 3, and 7. This result is contrary to previously reported results where 45S rDNA is presented only in three pairs of chromosomes [22,25]. Type III signals were located basically at the centromeric region of all chromosomes and from the proximal to middle part of the interstitial region of long arm of chromosomes 1 and 2. Type IV signals were present in the terminal regions of both arms except for the short arm of chromosome 4 and the long arm of chromosomes 6 and 7.

To summarize, it is clear that the data from the various experiments, with regard to the B10v3 genome, mostly overlap. The bioinformatics analyses carried out are also consistent with information previously obtained by laboratory methods, confirming the effectiveness of in silico methods in comparing and aligning contigs in genomes. Although there are still imperfections in these tools, future improvements should be made in terms of usability, increased batch data, and speed. Nevertheless, we were able to organize data for the B10v3 genome using RagTag for the two cucumber genomes: 9930 and Gy14. Although the genomes are not 100% assembled, we can see a huge progression in assembling and describing the structures present on the chromosomes.

Despite the existence of gaps between contigs on each genome, it is possible to establish a concordant match between contigs and chromosomes; however, the results of assigning B10v3 contigs to chromosomes depend on the reference genome used (line 9930 or Gy14).

Assembling full length genomes are a challenge for the future. For this purpose, techniques are needed to sequence the longest possible molecules containing accurate information about repetitive elements, which can then be correctly sequenced. At the moment, as can be seen from the FISH analyses performed and described here, the repetitive elements are located on chromosomes but, for the most part, their detection in existing assemblies in the context of assignment to chromosomes proves impossible. In contrast, areas containing genes are fairly well described. Both the possibility of sequencing techniques and sequence data processing programs need to be developed to acquire the knowledge of complete sequences.

We presented an established data integration pipeline that integrates experimental data with in silico data from multiple previous analyses, providing a clear picture of localization on the B10v3 cucumber genome. The alignment of contigs on chromosomes provides a new starting point for further comparative analyses and also enables the localization of genes on chromosomes in resequencing analyses, making it possible to detect their hotspots, which has not been possible so far for the B10v3 genome. Furthermore, the information obtained on the similarity of the B10v3 genome to the 9930 and Gy14 genomes provides additional knowledge on the European version of cucumber and brings us closer to a more complete cucumber pangenome.

The above work is particularly relevant for teams working with cucumber species. However, we would like to emphasize that the analysis scheme presented can be used to correlate data in other species. Currently, there are many centers working on genomes of the same species, and it is useful to be able to easily exchange and correlate data. This will facilitate work, especially in terms of detailed structural and functional data.

## 3. Materials and Methods

### 3.1. Sample Data Curation

In subsequent analyses, the FASTA files and annotations for the B10v3, 9930v3, and Gy14v2.1 cucumber genomes from the CuGenDBv2 database [26] were used.

The retrieved genomes differ in assembly characteristics, so the first step was to prepare a summary for each genome using the sequence-stats program [27]. At this stage, the size of each genome, number of contigs, percentage of bases, N50 value, maximum, minimum, and median contig size were calculated.

Then, multiple sequence alignments of the B10v3, 9930, and Gy14 genomes were prepared together with visualization in the Mauve [28] software. For its execution, FASTA sequences downloaded from the CuGenDBv2 database were used, and the progressive Mauve algorithm was applied.

### 3.2. Data Arrangement and Chromosome Assignment

In the next step, a contig rearrangement methods based on homology to reference genomes were used. In this case, contigs from the B10 genome were ordered, by comparison to the 9930 and Gy14 reference genomes, respectively, using RagTag [29], Mauve Contig Mover [28], and Multi-CSAR [30] programs. In order to position the contigs relative to the reference chromosomes, the 9930 and Gy14 reference genomes were modified by removing the extra scaffold sequences (not organized into chromosome 1–7), leaving only the sequences of the assembled chromosomes. Thus, the contigs were arranged only to the complete chromosomes in the 9930 and Gy14 reference genomes. Mauve Contig Mover was set to generate the arrangement after 6 iterations of the program’s operation while Multi-CSAR and Ragtag’s scaffold module were used with default setup parameters.

### 3.3. Comparative Analysis of Coded Protein Sequences

Sequentially, a search for common proteins annotated in the B10v3, 9930, and Gy14 genomes was performed using the BLASTP [31] software. For this purpose, the longest proteins (per gene) from the B10v3 genome were used. Then, based on reference files from the CuGenDBv2 database describing the encoded proteins for the 9930 and Gy14 lines, sequence search databases were created using the makeblastdb tool [31]. Then, using the BLASTP program, the parsed protein sequences from B10v3 data set were searched, narrowing the results to the single best e-value result. After that, annotations about the function and ontology of the retrieved proteins were matched based on data available in the CuGenDBv2 database for the 9930 and Gy14 lines.

For proteins showing low sequence identity to Gy14 or 9930 proteins, a BLASTP was conducted to the NCBI [32] database NR with a narrowing down to the *Cucurbitaceae* family (taxon 3650). The results were then annotated accordingly the Entrez database API [33] in the Python programming language. For proteins that were retrieved with identity lower than 80 in any of the previous BLAST program searches, proteins were retrieved based on the homology for whole NR database, available on the NCBI server.

### 3.4. Integration of Published Data about Cucumber B10 Genome Assembly

For the integration of previously obtained data from genome characteristics studies we used (1) data from marker position on the B10v3 genome [4], (2) DArT-seq data [7], and (3) FISH data [6]. With reference to the loci of markers (1) mapped on the B10v3 genome [4], these data were compared with the in silico alignment results obtained herein by the RagTag software.

Regarding the DArT-seq data, the 717 sequences were mapped to the B10v3 genome using BLASTN software. The obtained results were integrated with information on the position of the contigs in the B10v3 genome and combined with information on the assignment of the loci to the chromosome identified in the DArT-seq experiment. The consensus assignment of a contig to a chromosome was then determined using the criterion of the highest number of matches.

In the FISH analysis described in the publication on NGS in cucumber (*Cucumis sativus* L.) [6], the 7 STCs (sequence-tagged connectors) derived from the BAC clones were located by BLASTN on the contigs of the B10v3 genome.

### 3.5. BAC-FISH Experiment Description

Root tip microscope slides were prepared according to previously described protocols [24,34]. Chromosome slide preparation was performed with the air-dry method.

The selected BAC DNA was extracted using HiSpeed Plasmid Kits, and whole BAC DNA was used for the labeling. A Nick translation kit was used to label insert fragments with tetramethyl-rhodamine-5-dUTP (Roche Diagnostics, Mannheim, Germany).

The polymerase chain reaction (PCR) was used to amplify 45S rDNA fragments, following the method of Sogin [35]. Types III and IV tandem repeats of cucumber were amplified following Helm and Hemleben [36] method.

The FISH was carried out with a modified method for direct labeling according to Tagashira et al. [24].

FISH signal photomicrographs were taken under an Olympus AX70 “Provis” (Olympus, Tokyo, Japan) microscope equipped with an Olympus DP50 digital camera. The image processing was performed using Photoshop (Adobe, Los Angeles, CA, USA) [37], pseudo-colored by increasing the color definition of signals based on differential coloration.

### 3.6. Data Consolidation

As the output of the data integration, the interactive HTML reports were created that can be searched, filtered, and exported to Excel and CSV files using dedicated buttons. It includes information on the length of contigs in the B10v3 genome, the results of contig arrangement in the RagTag software with use of the 9930 and Gy14 genomes as references, markers from the B10v3 genome assembly, and contig assignment to chromosomes obtained in DArTseq and FISH analysis. Reports were prepared using the DT library in the R programming language, creating separate files:-A file integrating RagTag results with BLASTP searches and data for markers used in B10v3 genome stacking together with results of DArT-seq and FISH analyses (Appendix A);-Files showing the results of the BLAST program (Appendix A);-A file showing the genes assigned to each contig (Appendix A).

## 4. Conclusions

In the post-genomic era, where the amount of sequenced information is increasing day by day and sequencing techniques and bioinformatics tools are developing rapidly, there is a need for a good description of the data obtained, as well as the correlations between them. Having a well-described reference for comparative genomics research is fundamental, and one must strive to find the best pathways for the analysis of such large data sets, both experimental and in silico, and their mutual correlation. In the present study, we have shown that the coding regions in the B10v3 genome in relation to the analyzed genomes coincide and can be located on chromosomes. Unfortunately, the assembly of the genomes is not complete due to problematic repetitive elements. The analyzed genomic data represent little polymorphism, which may be indicative of specific individual traits, but may also be indicative of the quality of the sequence data and assembly techniques of these genomes.

## Figures and Tables

**Figure 1 ijms-24-04011-f001:**
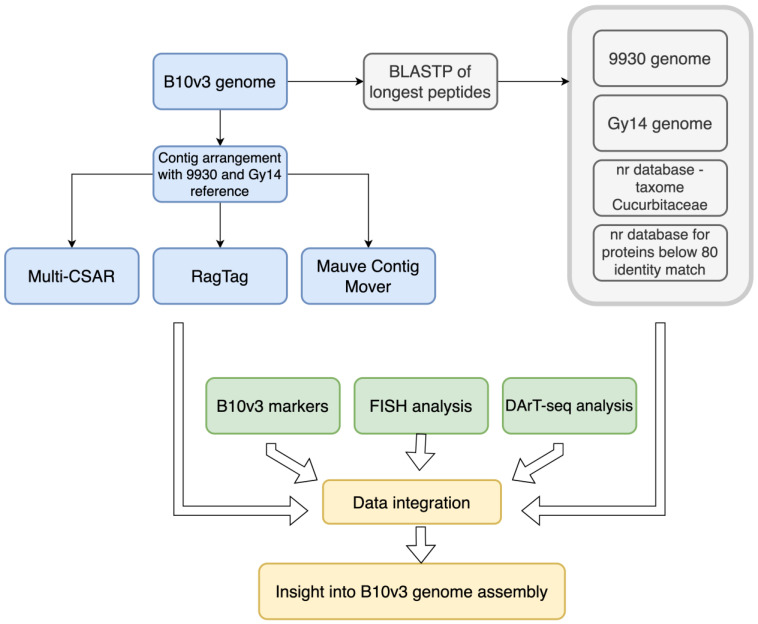
Pipeline of performed data consolidation.

**Figure 2 ijms-24-04011-f002:**
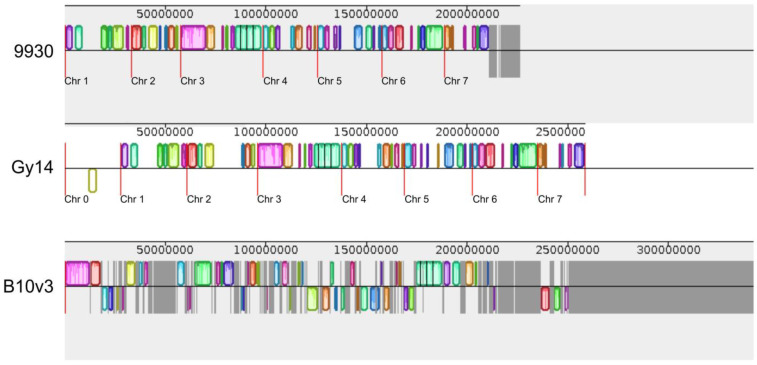
Multiple sequence alignment of 9930, Gy14, and B10v3 genomes of data from CuGenDBv2 database. The red vertical lines indicate the boundaries of the individual chromosomes, which have been signed according to the numbering in each genome. Colored boxes represent common elements between the compared genomes, and grey areas are unique to the depicted genome.

**Figure 3 ijms-24-04011-f003:**
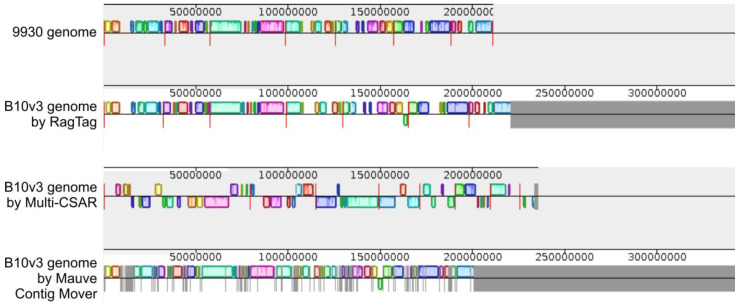
Comparison of arrangement tools: RagTag, Mauve Contig Mover, Multi-CSAR. As a reference 9930 genome was used. Colored boxes represent common elements between the compared genomes, and grey areas are unique to the depicted genome. The red vertical lines indicate the boundaries of the individual chromosomes.

**Figure 4 ijms-24-04011-f004:**
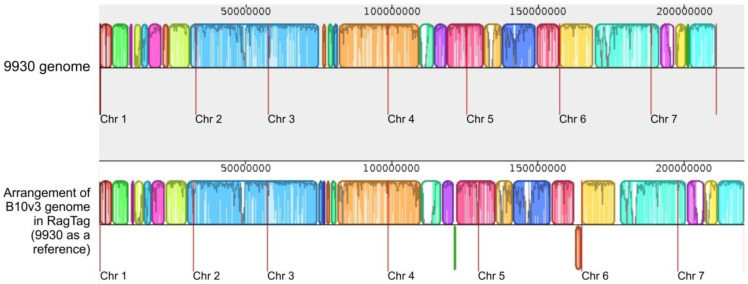
RagTag result of B10v3 genome rearrangement and assigned to chromosome with use of filtered 9930 reference genome visualized in Mauve software. The red vertical lines indicate the boundaries of the individual chromosomes.

**Figure 5 ijms-24-04011-f005:**
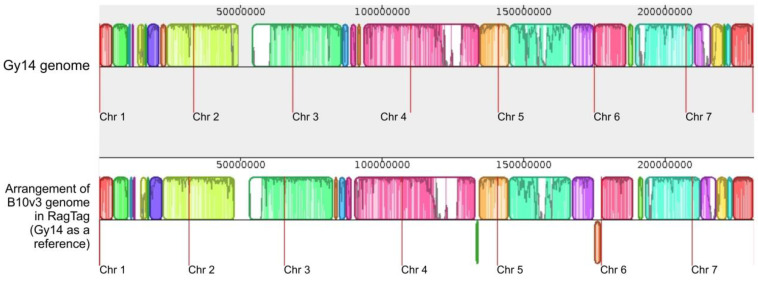
RagTag result of B10v3 rearrangement using filtered Gy14 reference and assigned to chromosome visualized in Mauve software. The red vertical lines indicate the boundaries of the individual chromosomes.

**Figure 6 ijms-24-04011-f006:**
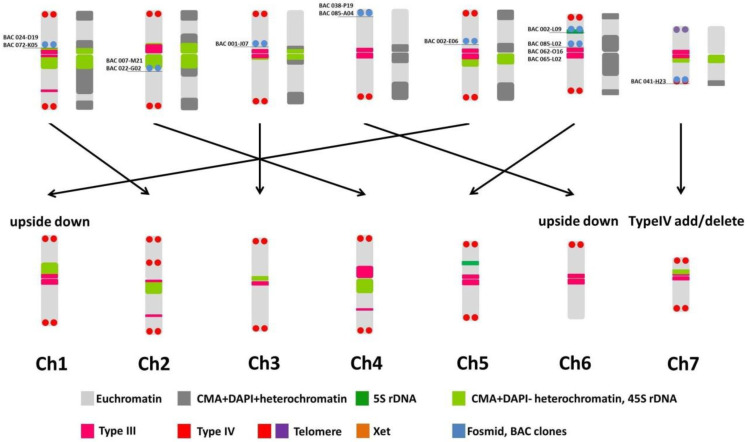
Comparative and corresponding B10 numbering (**upper**) to 9930/Gy14 numbering (**down**). Chromosome number (Ch) according to Han et al. [21] and Yang et al. [3] work. Karyotype of B10 showed FISH signals (**left**) and sequential DAPI and CMA staining (**right**). Karyotype of 9930 and Gy14 showed only FISH signals. The arrows indicate the chromosome equivalent of the B10v3 version (**upper**) vs. 9930 and Gy14 (**down**).

**Figure 7 ijms-24-04011-f007:**
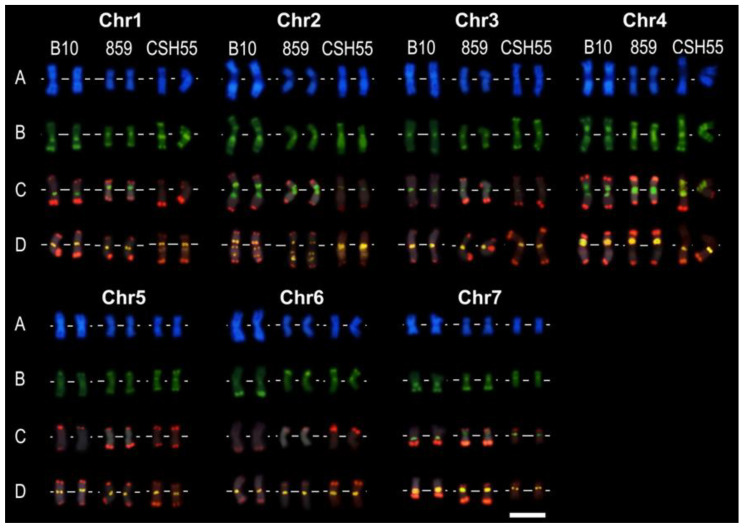
Probe localization on three cucumber line chromosomes: B10, 859, and CSH55. (**A**) DAPI staining, (**B**) CMA staining, (**C**) Type IV (red) and 45S rDNA (green), and (**D**) Type III (orange) and Type IV (red). Scale bar = 5 μm.

**Table 1 ijms-24-04011-t001:** Statistics of genomes B10, Gy14, and 9930 generated by sequence-stats program with calculated max, min, and median contigs lengths.

Sample ID	Genome	Contigs	GC (%)	A (%)	T (%)	G (%)	C (%)	N50	Max	Min	Median
B10v3	3.4 × 10^8^	8035	35.72	32.11	32.15	17.87	17.84	857,970	12,672,375	3684	15,301
Gy14	2.6 × 10^8^	8	30.01	29.86	29.78	15.00	15.00	33,288,019	41,698,299	23,759,298	32,825,499
9930	2.3 × 10^8^	85	32.81	33.60	33.56	16.40	16.40	31,125,843	40,877,379	5476	160,015

**Table 2 ijms-24-04011-t002:** Comparison of BLASTP search results in given sequence identity ranges. As an input file 16,104 proteins from B10v3 data were used. The individual rows of the table show the number of proteins found by the BLASTP software within the identity ranges for searches in databases created from the cucumber 9930, Gy14 proteomes, and an NR database narrowed to taxon 3650. The last row shows the sum of all proteins found for each BLAST search.

	Number of Found Proteins in BLAST Databases		
Identity Search Criteria	9930_db	Gy14_db	Nr Database Narrowed to 3650 Taxon
=100	9519	7422	7311
<100 AND ≥95	5008	4635	6002
<95 AND ≥90	579	1335	1488
<90 AND ≥80	315	906	718
<80	671	1788	404
Sum of found protein sequences	16,092	16,086	15,923

**Table 3 ijms-24-04011-t003:** Result of a BLASTN search for STC sequences in the B10v3 genome.

STC Sequence Identifier	Found Contig with BLASTN Program	Bioinformatics Consensus Result	Chromosome According to BAC Analysis Results
**STC1_Bam_024_D19_M13**	ctg105	Chr2	Chr2
**STC1_Bam_007_M21_M13**	ctg184	Chr4	Chr4
**STC1_Bam_022_G02_M13**	ctg184	Chr4	Chr4
**STC1_Bam_001_J07_M13**	ctg184	Chr4	Chr3
**STC1_Bam_065_A04_M13**	ctg1673	Chr5	Chr5
**STC1_Bam_002_E06_M13**	ctg2607	Chr3	Chr1
**STC1_Bam_041_H23_M13**	ctg1047	Chr7	Chr7
**STC1_Bam_002_L09_M13**	ctg197	Chr5	Chr5
**STC1_Bam_062_O16_M13**	ctg775	Chr5	Chr5
**STC1_Bam_072_K05_M13**	ctg105	Chr2	Chr2
**STC1_Bam_038_P19_M13**	ctg1000	Chr6	Chr6
**STC1_Bam_065_L02_M13**	ctg775	Chr5	Chr5

**Table 4 ijms-24-04011-t004:** A snippet of the final table from the HTML report showing the integrated data for 6 selected contigs detected by FISH analysis.

Contig Name	Chromosome B10v3 Markers	Chromosome 9930 RagTag	Chromosome Gy14 RagTag	Chromosome Blast 9930db	Chromosome Blast gy14db	Chromosome According to DArT-Seq Analysis	Chromosome According to BAC Analysis
ctg2607	Chr3	Chr3_9930	Chr3_Gy14	Chr3	Chr3	Chr3	Chr1
ctg105	Chr2	Chr2_9930	Chr2_Gy14	Chr2	Chr2	Chr2	Chr2
ctg184	Chr4	Chr4_9930	Chr4_Gy14	Chr4	Chr4	Chr4	Chr4
ctg197	Chr5	Chr5_9930	Chr5_Gy14	Chr5	Chr5	Chr5	Chr5
ctg1673	Chr5	Chr5_9930	Chr5_Gy14	Chr5	Chr5	Chr5	Chr5
ctg1047	Chr7	Chr7_9930	Chr7_Gy14	Chr7	Chr7		Chr7

## Data Availability

Publicly available datasets were analyzed in this study, according to cited articles.

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
