# Peer review of "Insight into the Organization of the B10v3 Cucumber Genome by Integration of Biological and Bioinformatic Data"

_ijms, 2023, doi:10.3390/ijms24044011_

Round 1
Reviewer 1 Report
This paper deals with the comparative analysis of the cucumber genomes, B10v3, 9930 and Gy14. The authors tackled the question of the similarities and differences in the coding protein sequences among the three genomes. I think the comparative analysis is very important for the related researchers to perform molecular functional genetic experiments. I have a few minor points.
In abstract, I am glad if short explanations regarding the 9930 and Gy1416 cucumber genomes are added. In other words, what are the differences between the B10v3 and their genomes?
Please describe the content of Table 2.
Author Response
Comment 1
This paper deals with the comparative analysis of the cucumber genomes, B10v3, 9930 and Gy14. The authors tackled the question of the similarities and differences in the coding protein sequences among the three genomes. I think the comparative analysis is very important for the related researchers to perform molecular functional genetic experiments. I have a few minor points.
Response:
Thank you very much for your review and your comments on the work.
Comment 2
In abstract, I am glad if short explanations regarding the 9930 and Gy1416 cucumber genomes are added. In other words, what are the differences between the B10v3 and their genomes?
Response:
Thank you very much for your review and your comments on the work. We added in the abstract part information about the differences about genomes which is mostly the origin of lines.
Comment 3
Please describe the content of Table 2.
Response:
Thank you for the suggestion, a description has been added for table 2. in the table caption, we also add description of table 2 in the text in Results part.
Reviewer 2 Report
1. What is the main question addressed by the research?
This paper has the following parts:
“The resulting pan-genome combines information on cucumber genomes assembled only at the chromosome level, indicating the need for a well-described reference both at the structural level of identified genes and at a larger structural level such as chromosomes.” and “In this work we present a new pipeline for comparative genomic analyses integrating high-throughput sequence data with biological data, resulting in the consolidation and structuring of knowledge on genomic draft structure” I think the impact is weak.
2. Do you consider the topic original or relevant in the field? Does it address a specific gap in the field?
I think the author wants to claim originality about the analysis shown in Figure 1.
3. What does it add to the subject area compared with other published material?
This paper is a kind of case report as it is a technical study. I think it would be better to add a comparison with B10v2.
4. What specific improvements should the authors consider regarding the methodology? What further controls should be considered?
If I may add, I should mention the superiority of the findings from this analysis.
5. Are the conclusions consistent with the evidence and arguments presented and do they address the main question posed?
I think the conclusions are consistent with those presented.
6. Are the references appropriate?
I think it's appropriate.
7. Please include any additional comments on the tables and figures.
I think it would be better to explain the multiple alignment results in Figures 2 and 3 in a little more detail.
The meaning of common parts should be explained.
I don't think there is any point in introducing TagTag.
I checked your manuscript and described comments below.
Cucumber is an important food eaten all over the world.
This paper provides a very good study of the B10v3 cucumber genome using bioinformatic methods and FISH.
I don't think I learned anything new, but I think it's important in that it shows how to do the analysis.
I have a suggestion. I think Ref.16 should be included in the text.
I don't think this paper has any major mistakes or grammatical problems.
Author Response
Comment 1
Comments and Suggestions for Authors
- What is the main question addressed by the research?
This paper has the following parts:
“The resulting pan-genome combines information on cucumber genomes assembled only at the chromosome level, indicating the need for a well-described reference both at the structural level of identified genes and at a larger structural level such as chromosomes.” and “In this work we present a new pipeline for comparative genomic analyses integrating high-throughput sequence data with biological data, resulting in the consolidation and structuring of knowledge on genomic draft structure” I think the impact is weak.
Response:
Thank you for your comment. We are in agreement with the Reviewer that we have been too vague about the impact and significance of our research. We have developed this issue in the introduction and discussion. We hope that there is no longer any doubt that the analyses that have been carried out are significant. Having a genome that is well characterised provides a reference point for performing subsequent analyses. Comparative analyses as well as the creation of the pangenome require genomes that have information on the best possible arrangement of structures, preferably at the chromosome level. The B10v3 genome, despite sequencing with PacBio long-read technology, has little information on the alignment of contigs to chromosomes, so the analysis and data integration performed provided significantly more information on this topic. The correct alignment of the contigs on the chromosomes provides a new starting point for further comparative analyses and can also enable additional studies at the cucumber pangenome level.
In the introduction, we have added a section showing the relevance of genome ordering for subsequent pangenome creation, together with an example of a publication in which the creation of a pangenome for Aspergillus fumigatus is shown after prior bioinformatics processing of sequencing reads.
Comment 2
- Do you consider the topic original or relevant in the field? Does it address a specific gap in the field?
I think the author wants to claim originality about the analysis shown in Figure 1.
Response:
Progress in crop genomics requires developing efficient ways to process, characterise and compare data at different omic levels. This is possible through the development of systems and software that can do high-throughput genetic data processing.
Here, we present three tools used to compare the cucumber genome and outline their advantages and disadvantages. To establish the contig order of the B10v3 reference genome at the chromosome scale, we were able to develop and identify a set of steps and automated tools presented on Fig1. This work outlines strategies for rapidly expanding genetic systems and genomic resources in cucumber and provides guidelines for applying these to other plants.
Comment 3
- What does it add to the subject area compared with other published material?
This paper is a kind of case report as it is a technical study. I think it would be better to add a comparison with B10v2.
Response:
The manuscript presented relates to integrating experimental, genomic and bioinformatics data. One of the most important results of the work presented is the comparison of the B10v3 genome with the Chinese 9930 and US Gy14 genomes. We have deliberately not made a comparison of the B10v3 to the B10v2 version in this work, as such a comparison has already been described in a publication describing the origin of the B10v3 genome - Osipowski et al. 2020, where detailed statistics for the contigs of B10v2 (long read corrected draft), and B10v3 (Illumina-corrected draft) are presented.
Comment 4
- What specific improvements should the authors consider regarding the methodology? What further controls should be considered?
If I may add, I should mention the superiority of the findings from this analysis.
Response:
Thank you for your comment and suggestions. In the introduction and in the discussion, we have added a section highlighting the importance of the analysis carried out and the potential usefulness of the results obtained.
Comment 5
- Are the conclusions consistent with the evidence and arguments presented and do they address the main question posed?
I think the conclusions are consistent with those presented.
Response:
Thank you for your comment.
Comment 6
- Are the references appropriate?
I think it's appropriate.
Response:
Thank you very much for this comment, the whole text has been checked.
Comment 7
- Please include any additional comments on the tables and figures.
I think it would be better to explain the multiple alignment results in Figures 2 and 3 in a little more detail.
The meaning of common parts should be explained.
I don't think there is any point in introducing TagTag.
Response:
Thank you very much for this comment, the Figures: 1, 2, 3, 4 and 5 have been changed in order to better exlplain the results and and highlight the starting point for using the RAgTag programme
Comment 8
I checked your manuscript and described comments below.
Cucumber is an important food eaten all over the world. This paper provides a very good study of the B10v3 cucumber genome using bioinformatic methods and FISH.
I don't think I learned anything new, but I think it's important in that it shows how to do the analysis.
I have a suggestion. I think Ref.16 should be included in the text.
Response:
Thank you very much for this comment, we checked again the Ref. 16.
Comment 9
I don't think this paper has any major mistakes or grammatical problems.
Response:
Thank you. We appreciate your support.
Reviewer 3 Report
Authors present interesting research related to sequencing and assembly of the B10v3 cucumber genome based on two other already sequenced cucumber genomes 9930 and Gy14.
Study is well planned and performed, results support conclusions.
Minor comments:
1. Latin names of plant write in italics- check the entire text.
2. Describe in the introduction section if related approach combining experimental results of related genomes and similar bioinformatic strategy was used in other studies.
Author Response
Comment 1
Comments and Suggestions for Authors
Authors present interesting research related to sequencing and assembly of the B10v3 cucumber genome based on two other already sequenced cucumber genomes 9930 and Gy14.
Study is well planned and performed, results support conclusions.
Response:
Thank you very much for these comments, we have revised the manuscript as suggested by the reviewer.
Comment 2
Minor comments:
- Latin names of plant write in italics- check the entire text.
Response:
Thank you very much for this comment, the whole text has been checked.
Comment 3
- Describe in the introduction section if related approach combining experimental results of related genomes and similar bioinformatic strategy was used in other studies.
Response:
Thank you for this comment and suggestion. A section has been added to the introduction referring to a review publication listing the methods currently used to map plant genomes, and a section highlighting the essence of the analysis carried out. A whole branch of bioinformatics tools enabling the appropriate de-novo ordering of reads after sequencing is currently being extensively developed. Programs based on a selected reference genome, such as RagTag, are one method for this purpose. While these programmes were designed to work by default with post-sequencing assambled reads, in the analysis we conducted, the already well described contigs of B10v3 genome were used. When the B10v3 genome was sequenced, contigs were assembled from reads derived from sequencing using PacBio long read technology. The assembly of contigs derived from such sequencing accounts for their appropriate quality. Our analysis used assembled contigs by ordering and localising them on chromosomes. This type of analysis using RagTag (published in 2022) and the integration of the results of laboratory experiments for the B10v3 genome has not been described before and represents an approach to organise a genome that has already been assembled and described, but which for various reasons has not been arranged at the level of chromosomal organisation.